# Multifunctional Freestanding Microprobes for Potential Biological Applications

**DOI:** 10.3390/s19102328

**Published:** 2019-05-20

**Authors:** Nana Yang, Zhenhai Wang, Jingjing Xu, Lijiang Gui, Zhiqiang Tang, Yuqi Zhang, Ming Yi, Shuanglin Yue, Shengyong Xu

**Affiliations:** 1Key Laboratory for the Physics & Chemistry of Nanodevices and Department of Electronics, Peking University, Beijing 100871, China; yangnana@pku.edu.cn (N.Y.); wangzhenhai@pku.edu.cn (Z.W.); jj_xu1990@163.com (J.X.); tangzhiqiang@pku.edu.cn (Z.T.); 2School of Microelectronics, Shandong University, Jinan 250100, China; 3Department of Micro-Nano Fabrication Technology, Institute of Electrical Engineering, Chinese Academy of Sciences, Beijing 100190, China; lijianggui@mail.iee.ac.cn; 4Neuroscience Research Institute and Department of Neurobiology, School of Basic Medical Sciences, Peking University, Beijing 100083, China; zhangyuqi960418@163.com (Y.Z.); mingyi@bjmu.edu.cn (M.Y.)

**Keywords:** freestanding micro-probe, stainless steel, thermocouple, electrode, biological application

## Abstract

Deep-level sensors for detecting the local temperatures of inner organs and tissues of an animal are rarely reported. In this paper, we present a method to fabricate multifunctional micro-probes with standard cleanroom procedures, using a piece of stainless-steel foil as the substrate. On each of the as-fabricated micro-probes, arrays of thermocouples made of Pd–Cr thin-film stripes with reliable thermal sensing functions were built, together with Pd electrode openings for detecting electrical signals. The as-fabricated sword-shaped freestanding microprobes with length up to 30 mm showed excellent mechanical strength and elastic properties when they were inserted into the brain and muscle tissues of live rats, as well as suitable electrochemical properties and, therefore, are promising for potential biological applications.

## 1. Introduction

The fabrication of multifunctional smart microprobes is a hot topic and a technical challenge. Smart microprobes are expected to play increasingly important roles in many fields of fundamental biology such as neurosciences, brain–computer interfaces, and a variety of clinic applications. Microprobes could be roughly cataloged into two major classes: flexible probes and rigid probes. In the last two decades, many kinds of flexible probes made with flexible materials were reported and were applied to solve many problems in science research and medical application [1,2,3,4,5,6,7,8,9,10,11,12,13,14,15,16,17,18,19,20,21]. However, in many cases, such as deep brain stimulation (DBS), rigid and long probes are required. 

The development of smart rigid probes could be traced back to the 1970s. Miniature-size neural microprobes were fabricated by several groups, offering advanced capabilities such as integrated electronic circuitry, precise control over the spatial distribution of electrode sites, and high density of electrodes [22,23,24,25,26,27,28]. However, most of the devices were designed for intra-cortical applications, which limited the length of the microprobes to a few millimeters, making them not suitable for detecting thermal and electrical properties in deep tissues of animals. The encapsulated metal wires introduced for single-site recording are the simplest and earliest type of devices that can be constructed with lengths in excess of several centimeters [29]. Carlos M. Florez-Quenguan et al. made a three-dimensional (3D) microprobe for DBS. The main component of the multisite 3D microprobe is a flexible planar microprobe wrapped around or assembled on a cylindrical support structure [30]. In another approach by G. Stavrinidis et al., based on microfabrication techniques, a thick electroplated film was employed as the rigid structural layer [31]. Recently, various smart 3D probes were developed [32,33,34,35,36,37,38,39] to detect stress, stimulate visual neurons, monitor bone disease, and record neuron activities and thermal effects.

In most current rigid 3D microprobes, the substrate materials were Si [28,35,37,40,41,42], glass [29,34,35,43,44], or plastic materials such as SU-8 [45,46,47] and polyethylene glycol (PEG) [48,49,50,51]. Current metal-based probes for DBS have diameters of 1.2–2.0 mm. Such a large diameter may cause severe damage of brain tissues during implantation surgery. The goal of this work was to fabricate micro-nano-sensors on the body of stainless-steel (s.s.) needles, obtaining a multifunctional smart 3D microprobe for potential biological applications, with a diameter smaller than that of the current DBS probes. Compared to plastics, s.s. material has much better mechanical strength and elasticity [52,53,54,55,56,57,58]. It has a high modulus of elasticity around 195 GPa, which is higher than that of copper (~100–130 GPa) and titanium (~100 GPa) and much higher than that of gold (~14 GPa), aluminum (~0.3–70 GPa), silver (~7 GPa), and plastics (usually <10 GPa, e.g., SU-8, ~5 GPa [57], PEG, ~3 GPa [58]). In addition, s.s. is a “biocompatible” metal to human beings. For example, we have quite a large amount of iron (Fe) atoms in our bodies, especially in our red blood cells. In acupuncture therapy, which has been practiced for thousands of years, steel needles can be inserted into muscle tissues and stay there for a long time without causing inflammation, poisoning, or rejection reactions. These unique properties make s.s. a good candidate for supporting the frames of deep probes in clinical practices and fundamental biological research. In this study, we designed and fabricated a novel freestanding microprobe that we hope to apply to measure organ temperatures as well as the exact 3D distribution of local temperatures in a live animal under various conditions.

## 2. Materials and Methods

### 2.1. Stainless-Steel Substrate 

The probes were fabricated on circular SUS 304 s.s. substrates with the diameter of 10.0 cm and thickness of 150 µm. The substrates were mechanically polished with a shining, plain surface. Each s.s. substrate was also chemically cleaned in three steps before lithography processes. The first step was an ultrasonic cleaning in acetone for 15 min followed by washing in ethanol for 15 min and by rinsing in deionized water; the substrates were then blown dry with a nitrogen gun. Finally, each substrate was placed on a hot plate and baked at 95 °C for 30 min to completely dry its surface. 

### 2.2. Design of 3D Probes

Figure 1a shows the overall design of the s.s.-based multifunctional probes. On each 4 in. s.s. wafer, seven sword-shaped probes with varied width and length were patterned. Different widths (w) and lengths (L) were designed to meet different requirements in measuring local temperatures at deep levels and in tissues with varied toughness in animal experiments. The configurations of the w and L parameters were 200 µm and 10.0 mm, 300 µm and 15.0 mm, 400 µm and 20.0 mm, and 500 µm and 30.0 mm. L was defined as the length of the probe from tip to handle, and w was defined as the width close to the probe tip. It should be mentioned that the width of the probe gradually increased from tip to handle, as shown in Figure 1b. 

Figure 1b schematically presents the major structure of a freestanding micro-probe with a built-in sensor array of thin-film thermocouple (TFTC) for local temperature measurement and electrodes for electrical signal detection. Each probe consisted of two main parts: a probe body and a connection plate. To ensure that each probe had sufficient mechanical strength, a sword shape was designed with triangular tips and a thickness (t) of 150 µm for all probes. 

For the 200 µm-wide probe, two TFTCs and two electrodes were designed. For the 500 µm-wide probes, the number of TFTCs and electrodes was increased to 4–6. To avoid reactions with biological tissue fluids, the TFTC array was covered with a thin oxide layer, while the electrodes were left open at the testing zone. 

The connection plate consisted of a triangle with a base length of 1.0 cm and a rectangle with a size of 5.0 mm × 1.0 mm. Its main function was to connect the leads of the TFTC array to the test system. 

### 2.3. Fabrication of Freestanding Microprobes

Figure 2a–h illustrates some major technical processes performed in a standard class-1000 cleanroom for the fabrication of the freestanding microprobes. These complicated processes involved two main steps: (1) defining the shape and position of the probe and fabricating the TFTC array and electrodes; (2) obtaining freestanding microprobes through wet etching. 

First, the as-purchased s.s. substrate was cleaned thoroughly as described in Section 2.1 (Figure 2a). Then the substrate was coated with a 3 µm-thick negative photoresist layer of SU-8 (2005, Microchem, Westborough, MA, USA) using a benchtop spin coater (CEEVR 200X, Brewer Science, 4000 rpm, 60 s) and was patterned by photo lithography on a mask aligner (Karl Suss MJB4, Suss, Germany) to define the shapes of the microprobe sword and connection plate (Figure 2b). For this step, soft-baking (95 °C 2.0 min), exposure (to a UV dose of 90 mJ/cm^2^), post-baking (95 °C, 1.0 min), developing (SU-8 Developer, Microchem, Westborough, MA, USA, 1.0 min), and hard-baking (150 °C, 30 min) were performed in sequence. These were the standard processing parameters for SU-8 2005 [59].

The SU-8 layer was used as an insulating layer to separate the TFTC array and electrodes from the s.s. substrate, and it also reduced the surface roughness thus ensuring a much higher yield of good sensors built on the microprobe. 

Next, positive photoresist AR-P 5350 (Allresist, Strausberg, Germany) was used to make TFTC patterns, following our previous procedures [60,61,62,63,64]. A 2 µm-thick AR-R 5350 was spin-coated on top of the front surface at 4000 rpm for 60 s and dried at 110 °C for 3 min. Patterns of TFTC array were carefully aligned with the Karl Suss MJB4 mask aligner to ensure they were located in the center of the probes defined with SU-8 patterns. 

After an open window for the deposition of Pd (or Cr) film was created, an oxygen plasma process was performed to remove residual photoresist (Ion Wave 10, PVA TePLa) before thin-film deposition. To improve the bonding strength of the Pd film on the SU-8 layer, a 5 nm-thick Ti thin film was deposited as an adhesion layer. Both Ti layer (5 nm) and Pd layer (thickness 90 nm) were deposited with an electron-beam evaporator (DE400, DE Tec, CN). The Ti–Pd stripe pattern appeared after the lift-off process in acetone (Figure 2c). The circular micro-electrodes for potential electrical measurements were also formed in this layer. A TFTC sensor was made of two different metals, i.e., palladium (Pd) and chromium (Cr) thin films, so the processes for alignment and formation of Pd and Cr patterns and deposition of Pd and Cr thin films were repeated twice. The Cr layer (120 nm) was deposited with a magnetron sputtering system (PVD75, Kurt J. Lesker, USA) in Ar. After lift-off, the TFTC arrays were completely built on the probes (Figure 2d). Then, a thin layer of HfO_2_ film (20 nm) was sputtered on the TFTC arrays for insulation, while still leaving the circular micro-electrodes testing regions (Figure 1b) open (Figure 2e). Finally, a 3 µm-thick SU-8 layer was patterned on the HfO_2_ layer to protect the devices from damage. Similarly, a SU-8 layer was spin-coated over the back side of the substrate for protection during etching (Figure 2f).

The wet-etching process was performed after the TFTC arrays and micro-electrodes were fabricated. The s.s. substrates were etched in an acidic solution of FeCl_3_, HCl, and HNO_3_, with a mixture of 40.0 g of FeCl_3_ powder, 16.0 mL of 37.0% HCl, and 16.0 mL of 97.2% HNO_3_ in 160 mL deionized H_2_O. A 10 µm-thick positive photoresist (SPR-220, Dow, USA) was used as the mask layer on the SU-8 layer to protect the existing TFTC devices in the wet-etching process. This mask layer was aligned and patterned on the device with standard photolithography processes. Soft-baking of the SPR-220 was done at 115 °C for 90 s (Figure 2g).

The etching was performed at 40 °C in a water bath via the floating technique [65,66]. The total etching period took around 65 min for the sword-shaped probes to thoroughly separate from the substrate. After several thin connecting bridges between each probe and the substrate base were cut with a sharp knife, a 3D freestanding microprobe was ready (Figure 2h). 

It should be mentioned that the 150 μm-thick s.s. substrate was not etched through a single process of 65 min. In order to ensure that the device structures of the TFTC and electrodes were not damaged in the etching process (due to over etching in the horizontal direction), after every 5–10 min, the wafer was taken out of the etchant, the SPR-220 photoresist was removed with acetone, and the wafer was rinsed and dried and then covered again with a new layer of SPR-220 photoresist. Alternately, the wafer could be processed after slightly longer time intervals (i.e., 10 min or more), but this increased the risk of breaking the TFTC patterns close to the sides of the probe. 

### 2.4. Device Testing

We calibrated the TFTC arrays on freestanding microprobes in an oil bath through a homemade calibration platform [63,66,67]. During the calibration process, we controlled the oil temperature by heating the hot plate and inserted the sword of the probe into the oil to reach the same temperature. The cold ends of the device were kept at room temperature. The difference in temperature between the hot end and cold end regions was recorded by two identical standard K-type thermocouples. The thermopower outputs of the TFTCs on the probe were measured with one nano-voltmeter (Keithley 2182A) via a multiplexer. The stress–strain measurements were performed on an Instron universal testing machine (Instron, USA). The electrochemical impedance spectroscopy (EIS) was measured with an electrochemical workstation (CHI660e, CH Instruments, USA) in 1x phosphate-buffered saline (PBS, pH 7.4) at room temperature with a three-electrode configuration, where the tested electrodes were working electrodes, while an Ag/AgCl electrode and a large-surface-area platinum wire served as reference and counter electrodes, respectively. During the test, frequency varied from 10 to 100 kHz.

Surface smoothness was investigated with an atomic force microscope (AFM, Dimension Icon, Bruker, Billerica, MA, USA). The morphology of probe and TFTCs was characterized with an optical camera (Axio Scope A1, Carl Zeiss, Oberkochen, Germany) and a scanning electron microscope (SEM, Quanta 600FEG, FEI, Brno, Czech Republic). Elastic measurement of the probes was performed with several pieces of silicon rubber. The sword tip of each probe being tested was mechanically pushed against the rubber surface. The sword part of the probe was bent, and the pushing force was then reduced to check recovery of the sword shape.

In addition, experiments on live animals were performed. Adult male Sprague–Dawley rats (300–350 g) were provided by the Department of Laboratory Animal Sciences, Peking University Health Science Center (Beijing, China). All experimental procedures were approved by the Animal Care and Use Committee of Peking University Health Science Center (LA2017131, approved on March 1, 2017) and were in accordance with the rules of the Declaration of Helsinki of 1975, revised in 2008. Details are given in Appendix A.

## 3. Results and Discussions

### 3.1. Improvement of Surface Smoothness after Coating SU-8

The surface of the as-purchased s.s. substrate, although mechanically polished, was rough. Figure 3a–c present typical AFM micrographs of three different regions of an s.s. substrate which underwent the cleaning processes and was ready for the lithography process. The surface was covered with scratch marks resulting from the polishing process with sandpaper. The absolute up and down fluctuation in the Z direction (perpendicular to the surface) was around 380–460 nm. For a scanning area of 50 μm × 50 μm, the root-mean-square roughness was around 40–44 nm. 

After being coated with a 3 µm-thick layer of SU-8, surface smoothness improved remarkably. As typically shown in Figure 3d–f, the absolute surface roughness was around 50–70 nm, a sevenfold improvement over the bare s.s. substrate. The root-mean-square roughness was around 4–6 nm for a scanning area of 50 μm × 50 μm, which was a 7–10-times improvement. This SU-8 buffer layer, therefore, not only served as a good insulating layer for the TFTC and micro-electrodes built on the probe but also ensured a better substrate for the following complicated lithography and thin-film deposition processes. 

### 3.2. Lithography and Etching Results

Figure 4 presents optical photographs of two devices. One was a whole 4 in. s.s. wafer before the etching process, where Pd–Cr TFTC arrays were fabricated, and the other was a whole wafer after the etching process, where seven microprobes were clearly defined. The narrow bonding bridges that connected the substrate wafer and the 3D probes are locally shown in the magnified images in Figure 4c,d. These connection bridges helped the probes maintain their original positions on the substrate throughout the fabrication process. After the bridges were cut away from the wafer, we obtained individual freestanding microprobes that were ready for testing. 

Figure 5 presents more details of the probes during and after the fabrication process. On each probe, the measurement zones were defined as a circular window in the SU-8 layer. Figure 5a shows optical images of the measurement zones and their locations on the probe. Figure 5b is a photo of a freestanding microprobe assembled in an electrical connection socket. Such a device was ready for practical measurements. Figure 5c is a SEM image of the measurement zone, where the junction of a Pd–Cr TFTC is seen in a circular window of SU-8. Figure 5d is a SEM image for the front part of a freestanding microprobe, where the sensors are seen in the middle of the probe, and the two sides are the etched edges of the s.s. substrate. Figure 5e,f presents SEM micrographs showing the tip and side-view of the sword-like s.s-based microprobes, including the corrosion profiles.

### 3.3. Sensor Calibration

The TFTCs on freestanding microprobes were calibrated in an oil bath. The outputs of thermopower for each individual Pd–Cr TFTC were measured first at varied temperature differences between its hot and cold ends. The sensitivity of the device was obtained from the slope of a linear fitting of the measured data. Figure 6 shows a typical set of calibration data and the linear fitting. For this device, the sensitivity was approximately 5.24 ± 0.15 µv/k.

### 3.4. A. C. Impedance

The interface impedance of the electrode–electrolyte system influenced the ability to record neural signals. Figure 7 shows one set of the obtained data. For a typical electrode with 1256 µm^2^ opening area, the measured low impedance was around 520 kΩ, and the phase was −79.9° at 1 kHz. At higher frequency, the impedance decreased gradually because of the capacitor effect.

### 3.5. Testing of Mechanical Strength

The as-fabricated 150 μm-thick freestanding microprobes showed excellent elasticity and mechanical strength. The stress required to bend the stainless-steel probe at 90° was found to be 8.2–95 Mpa, with a standard deviation of 10 MPa, for probes with different lengths. Figure 8 shows one set of the measurement results. The sword part of the probe could be repeatedly bent hundreds of times and still recover to its original shape. Figure 9 shows a testing case, where a probe mounted in its electrical connection socket was pushed against a piece of hard rubber, then the pushing force was released, and the probe returned to the initial straight shape (Appendix A). This case proved that stainless-steel probes have better elasticity than silicon-based [35] and glass-based [34] probes. 

The mechanical strength of the freestanding microprobes was also tested in live animals. They could be mechanically inserted into tissues of brain, abdomen, and thigh muscles to deep levels at 5–20 mm without causing the probe sword and the sensors on them to break. Moreover, the probes remained straight when they were pulled out of the tough muscle and skin tissues (Appendix A, Appendix A). These tests showed that our s.s. probes had excellent mechanical strength with such a thin thickness and performed better than any plastic-based probe [48].

## 4. Conclusions

In summary, we presented detailed procedures for the fabrication of a freestanding, multifunctional micro-probe made from 150 µm-thick s.s. sheets by using standard cleanroom techniques of photolithography, spin-coating, thin-film deposition, and wet-etching. An array of Pd–Cr TFTCs for temperature measurement and Pd electrodes for electrical measurement were fabricated on top of the sword-shaped s.s. substrate. The probes with length up to 30 mm showed excellent mechanical strength and elastic properties when they were inserted in brain, skin, and muscle tissues of live rats. Calibrations showed that the TFTCs worked well on freestanding microprobes. These novel probes are promising for the deep-level detection of 3D temperature distribution in organs and tissues of live animals and may also serve as deep-level sensors for recording electrical signals in neurons and muscles.

## Figures and Tables

**Figure 1 sensors-19-02328-f001:**
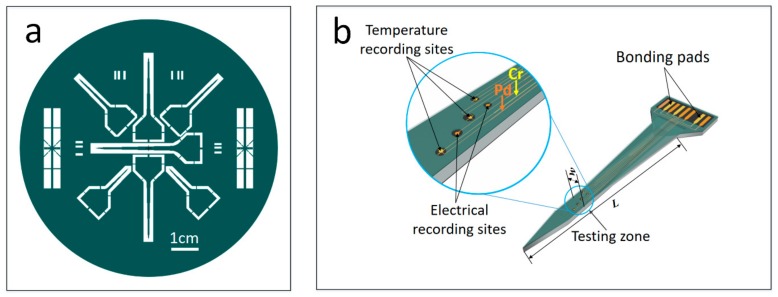
(**a**) Overall design of seven 3D micro-probes on a 4 in.-diameter s.s. substrate. (**b**) Schematic figure of the structure of a multifunctional 3D micro-probe.

**Figure 2 sensors-19-02328-f002:**
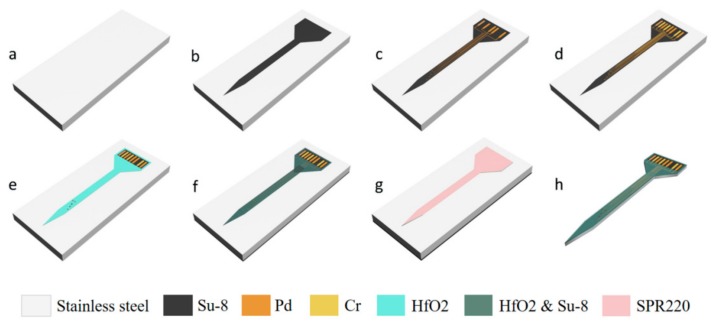
Illustration of the fabrication processes of a multifunctional 3D micro-probe. (**a**) Cleaned stainless-steel (s.s.) substrate. (**b**) Patterns of the probes with SU-8. (**c**) Ti–Pd patterns on the probes. (**d**) Cr patterns added to complete the Pd–Cr thin-film thermocouples (TFTCs) on the probes. (**e**) Covering with a HfO_2_ insulating layer, leaving testing opening windows. (**f**) Additional SU-8 layer to protect the TFTCs. (**g**) Coating of a mask layer of SPR-220 for the etching process. (**h**) The s.s. substrate was etched thoroughly after 65 min and cut from the substrate, resulting in a freestanding 3D probe.

**Figure 3 sensors-19-02328-f003:**
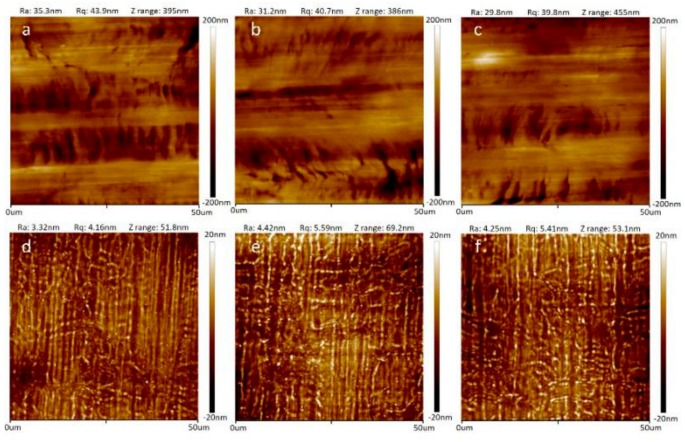
AFM micrographs of the substrate surfaces. (**a**)–(**c**) Results taken from a cleaned bare SUS 304 s.s. surface. (**d**)–(**f**) Results taken from the s.s. substrate covered with a 3 μm-thick SU-8 resist layer.

**Figure 4 sensors-19-02328-f004:**
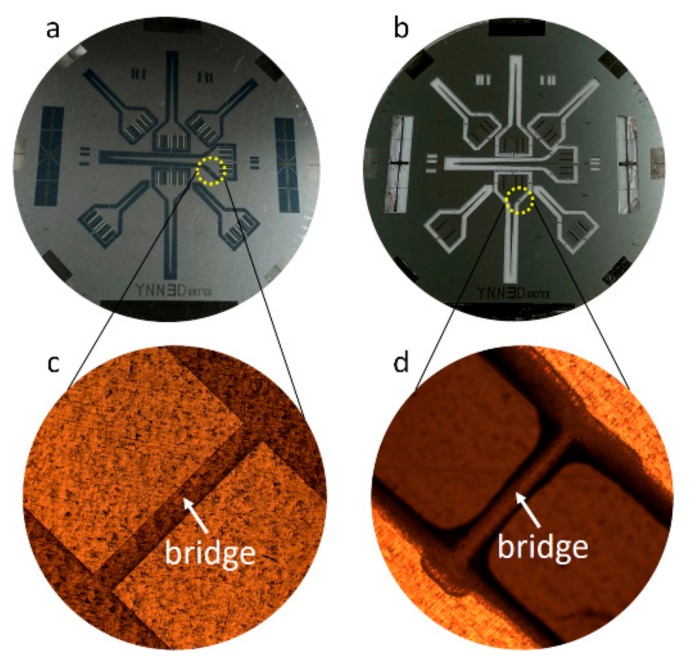
Optical photographs of the devices during the fabrication process. (**a**) A whole 4 in. substrate with fabricated TFTC arrays. (**b**) A whole substrate after the etching process. (**c**) An enlarged local area showing a narrow bonding bridge connecting the substrate and the 3D probe. (**d**) A close look at a bonding bridge after the etching process was completed, showing the probes still fixed in their original positions on the substrate.

**Figure 5 sensors-19-02328-f005:**
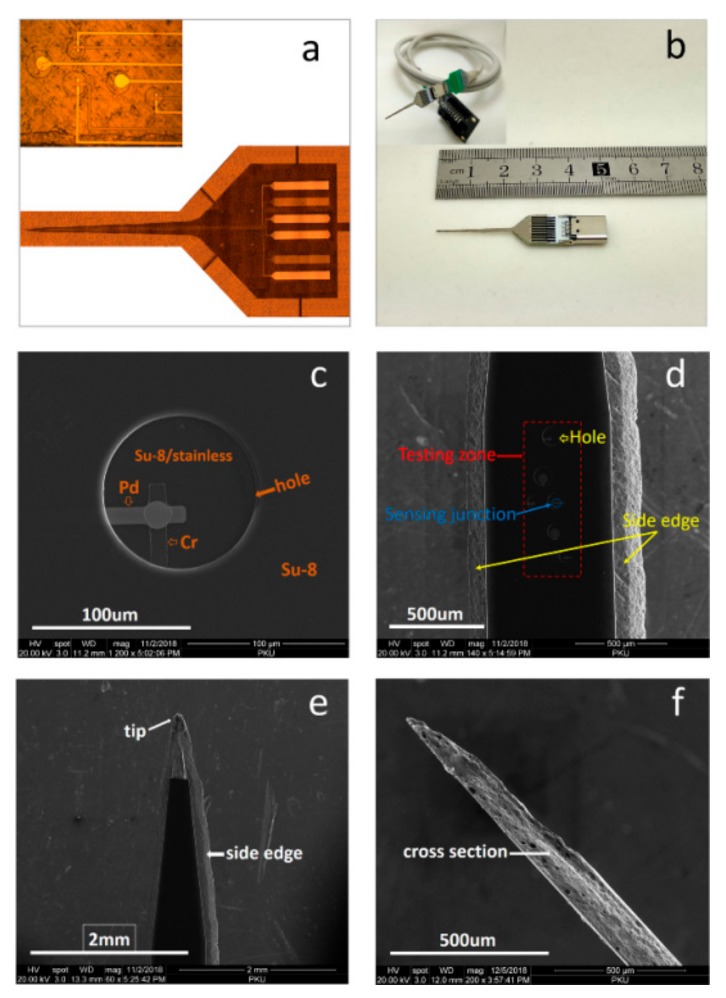
Optical images and scanning electron microscopy images of the as-fabricated device and assembled device. (**a**) Measurement zones on the probe before the etching process. (**b**) A microprobe assembled in an electrical connection socket. (**c**) SEM image of a measurement zone. (**d**) SEM image of the front part of a microprobe. (**e**) SEM micrograph of a probe tip. (**f**) SEM side-view of a probe tip.

**Figure 6 sensors-19-02328-f006:**
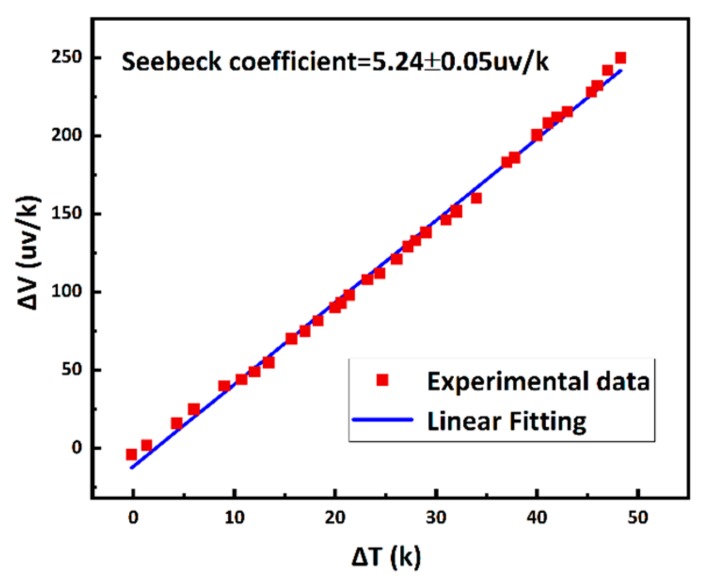
Calibration results for the Pd–Cr TFTC on a 3D microprobe.

**Figure 7 sensors-19-02328-f007:**
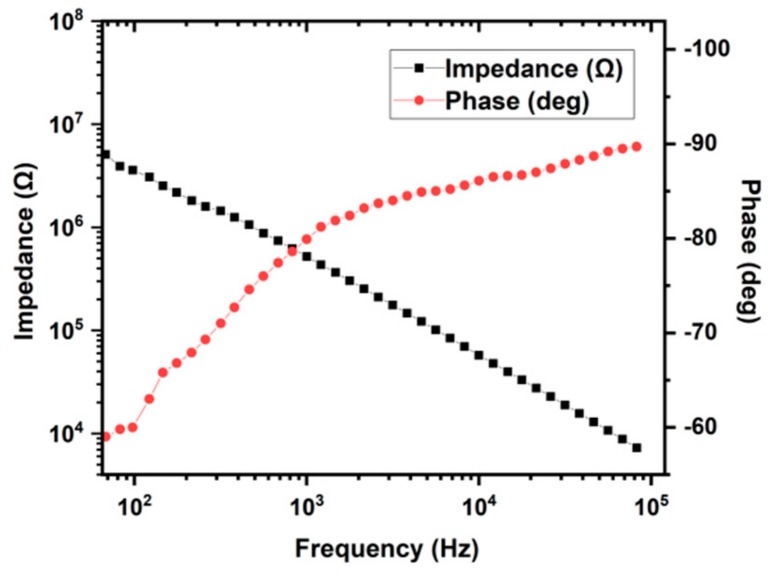
A typical electrochemical impedance spectroscopy of a Pd electrode.

**Figure 8 sensors-19-02328-f008:**
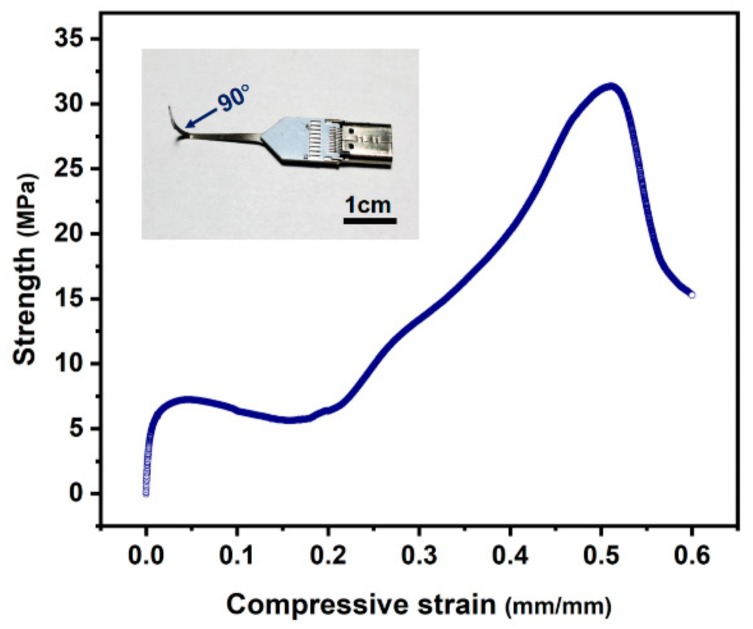
A typical stress–strain curve under compression for an as-fabricated 20 mm-long freestanding probe (shown in the inset).

**Figure 9 sensors-19-02328-f009:**
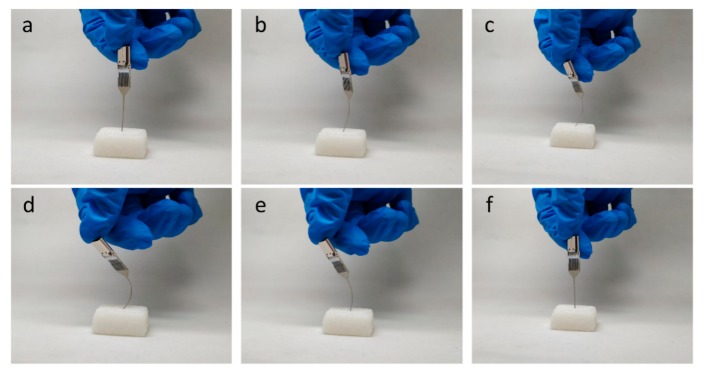
Photographs for the elasticity test of an as-fabricated microprobe with a hard rubber.

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
