# Peer review of "Multifunctional Freestanding Microprobes for Potential Biological Applications"

_sensors, 2019, doi:10.3390/s19102328_

Reviewer 1 Report
The authors present a passive probe for measuring temperature into tissue. The probe is manufactured from stainless steel . I do have a few questions though:
-       Introduction section: The authors discuss “soft” and “rigid” probes. I think it would be better to talk about flexible instead of soft probes. The probes aren’t really soft, they are flexible.
-       Introduction section: “green” material to humans --> I find this confusing. I assume the authors mean biocompatible?
-       Fig. 1b: It would be helpful to put the dimensions (length, thickness, height) on the figure. It is unclear high wide the probe is. The text mentions 200um and 500um?
-       Can the authors share insight into why they chose 150um thickness and 200/500um width? There are much smaller / thinner probes out there even as thin as 25um with a 70um width (in Si). (Jun et al, “Fully integrated silicon probes for high-density recording of neural activity”, Nature 2017 - https://www.nature.com/articles/nature24636). Why did the authors choose this fairly large probe design?
-       The authors call this a 3D probe. However, it is just a planar 2D structure. Since all the recording sensors are in a plane, this is clearly NOT a 3D probe, but a 2D probe. Please adjust the manuscript.
-       It is unclear why the authors chose to include thermocouple on the probe. Why is this a relevant parameter to sense with such a probe? And why are multiple sites needed? What is the relevant application?
-       The authors mention that TFTCs are covered with a thin oxide layer. How effective is this layer in avoiding reaction with tissue fluids? How long does it last? Has this been tested?
-       The authors mention they also include micro-electrodes. However, there is no further information on these electrodes. How big are they? What are the electrical properties (resistance/capacitance)? How stable are they over time? This is important to understand their usefulness for electrical recording.
-       The testing of the mechanical strength is inadequate. This test should have been done with a controlled force until breakage. Report the actual strength required to break the device. Please provide quantitative comparison with Si and glass probes.
-       While clearly the SS probes are stronger, is it actually needed? Could the probes not have been much thinner and still be strong enough to insert into tissue?
-       How flat are the probes and what is the maximum bending over the full length of 75px?
Author Response
Answers to the questions of Reviewer # 1
Q1. Introduction section: The authors discuss “soft” and “rigid” probes. I think it would be better to talk about flexible instead of soft probes. The probes aren’t really soft, they are flexible.
A1: Yes, “flexible” is better than “soft”. It has been corrected in the revised version.
Q2. Introduction section: “green” material to humans --> I find this confusing. I assume the authors mean biocompatible?
A2: Yes, we referred to “biocompatible”, and it has been modified in the revised version.
Q3. Fig. 1b: It would be helpful to put the dimensions (length, thickness, height) on the figure. It is unclear high wide the probe is. The text mentions 200um and 500um?
A3: In the revised version we have added in the dimensions (length, thickness, width) in Fig. 1b. The thickness (t) was the same for all probes, i.e., 150 µm. The widths (w) and lengths (L) of our as-fabricated probes had 4 kinds of (w, L) configurations, i.e., (200 µm, 10.0 mm), (300 µm, 15.0 mm), (400 µm, 20.0 mm) and (500 µm, 30.0 mm). However, the value for w was defined as the width close to the probe tip, and the probe width increased gradually as a sword does.
Q4. Can the authors share insight into why they chose 150um thickness and 200/500um width? There are much smaller / thinner probes out there even as thin as 25um with a 70um width (in Si). (Jun et al, “Fully integrated silicon probes for high-density recording of neural activity”, Nature 2017 - https://www.nature.com/articles/ nature24636). Why did the authors choose this fairly large probe design?
A4: The thickness and width of the probe are not the main concern of our project. The main concern is the length. We aimed at inserting such a long probe into organs of live animals to detect inner temperature distribution of an animal in different status. For instance, it is suggested that cancer cells are more active than normal cells so that a cancer tumour might have a higher local temperature than the rest part of the organ. This can be verified by measurement with our long probes. In another application, we plan to measure the inner temperature distribution of an animal in hibernation status, and a long micro-probe seems the only choice.
To maintain sufficient mechanical strength for a 3 cm long probe, and to build 5-10 sensors on the probe, we designed the width of 200-500 µm. 150 µm is the thinnest thickness of high quality s.s. sheet our supplier could currently deliver. Yes, we agree with the reviewer that both the thickness and width can be further reduced and optimized for particular applications.
In the revised version, we have added in this reference paper.
Q5. The authors call this a 3D probe. However, it is just a planar 2D structure. Since all the recording sensors are in a plane, this is clearly NOT a 3D probe, but a 2D probe. Please adjust the manuscript.
A5: To avoid confusing, in the revised version we have changed the term “3D probe” to “freestanding probe”. We used to work on physics of micro- and nano-sized materials and devices. A thickness of 150 µm is quite a big dimension, and is comparable to the width of 200-500 µm. Thus each probe is indeed a rigid three-dimensional device, with built-in sensors on the surface.
Q6. It is unclear why the authors chose to include thermocouple on the probe. Why is this a relevant parameter to sense with such a probe? And why are multiple sites needed? What is the relevant application?
A6: The main applications for this kind of long probes are to measure the exact three dimensional distribution of temperatures in a live animal under various conditions. For this purpose, current optical methods and other contact mode thermal meters seem not applicable. And, in order to simultaneously detect local temperatures at different depths of an organ, sensors are designed at multiple sites.
Q7. The authors mention that TFTCs are covered with a thin oxide layer. How effective is this layer in avoiding reaction with tissue fluids? How long does it last? Has this been tested?
A7: Yes, in a short time scale this thin oxide layer was sufficient to avoid bio-chemical reaction with fluids and cells in live tissues. The thin oxide layer protected the testing zone of the TFTC sensors, and the rest parts were covered with a 3 µm thick SU-8 layer, as described in the manuscript.
We have fabricated similar devices and applied them in detecting local thermal effects of cultured individual cells. Please refer to this work for details: “F. Yang, G. Li, J. M. Yang, Z. H. Wang, D. H. Han, F. J. Zheng, and S. Y. Xu*, “Measurement of local temperature increments induced by cultured HepG2 cells with micro-thermocouples in a thermally stabilized system”, Sci. Rep., 7 (2017) 1721”. In that application, arrays of TFTCs with similar structure and materials to this work were put in bio-liquids and live cells for days, and they worked very well.
On the other hand, the major application of the probes presented in this work is to detect the inner temperature of live (small) animals, and each time the measurement procedure lasts for only 10 s to 1-2 minutes, leaving a much shorter time for unexpected reactions.
Q8. The authors mention they also include micro-electrodes. However, there is no further information on these electrodes. How big are they? What are the electrical properties (resistance/capacitance)? How stable are they over time? This is important to understand their usefulness for electrical recording.
A8: As just mentioned, the main purpose of these probes is to measure deep level local temperatures of live animals. Electrodes are designed as additional choice for electrical measurements. The data of electrochemical impedance spectroscopy of the Pd electrodes have been given in the revised version. As shown in the following Figure, for a typical electrode with 1256µm2 opening area, the measured low impedance was around 520 kΩ, and the phase was -79.9° at 1 kHz. At higher frequency, the impedance decreased gradually because of capacitor effect. In addition, Detailed thermal and electrical measurements were ongoing, and it will take a long time to obtain valuable data.
Figure 1        A typical electrochemical impedance spectroscopy of a Pd electrode
Q9. The testing of the mechanical strength is inadequate. This test should have been done with a controlled force until breakage. Report the actual strength required to break the device. Please provide quantitative comparison with Si and glass probes.
A9: We appreciate the reviewer for this professional comment. We have performed precise mechanical measurements for the as-fabricated probes, and obtained their stress-strain curve under compression. The stress required to bend the stainless steel probe at 90 degrees was found to be 8.2-95 MPa, respectively, for probes with different length. The new data have been added in the revised version. The following figure shows one set of the measurement results.
Figure 2        A typical stress-strain curve under compression tested from an as-fabricated 20 mm long freestanding probe (shown in the inset).
Q10. While clearly the SS probes are stronger, is it actually needed? Could the probes not have been much thinner and still be strong enough to insert into tissue?
A10: Yes. Currently we found that s.s. substrate is the best choice.
We have been negotiating with the substrate supplier to offer thinner s.s. foil (say, < 100 µm) that they never fabricated before. From the current testing results, we believe that a thinner thickness of the probe will still be applicable for inserting into soft tissues (such as brain, liver, lung), but may fail in penetrating skin and muscle.
We will try to make smart sensors on natural materials such as long thorns of cactus, but that takes time.
Q11. How flat are the probes and what is the maximum bending over the full length of 75px?
A11: When the s.s. substrates were received, they looked plat. They were cut from large foils by machine and mechanically polished. After all the fabrication processes and when they were cut off from the substrate and put on a piece of glass, these probes still looked flat and straight. However, we have no facility to measure the flatness of each probe. For our applications of these probes, we needed to insert them into deep tissues of live animals (under anaesthetized state), so we didn’t pay attention to the flatness. Did we understand well the question of the Reviewer?

Reviewer 2 Report
The authors designed a freestanding, multifunctional micro-probe, which is including an array of Pd-Cr TFTCs for temperature measurement and Pd electrodes for electrical measurement. The probes showed excellent mechanical strength and elastic property when they were inserted in brain of live rats. But other than the mechanical property, I did not see any electrical multifunction as the authors proposed, such as electrochemical impedance spectroscopy of the Pd electrodes, temperature distribution in the rat’s brain, electrical signals of neurons, etc. So, I recommend the authors provide these details to prove their micro-probe’s function before the publication of manuscript in Sensors.
Author Response
Answers to the questions of Reviewer # 2
The authors designed a freestanding, multifunctional micro-probe, which is including an array of Pd-Cr TFTCs for temperature measurement and Pd electrodes for electrical measurement. The probes showed excellent mechanical strength and elastic property when they were inserted in brain of live rats. But other than the mechanical property, I did not see any electrical multifunction as the authors proposed, such as electrochemical impedance spectroscopy of the Pd electrodes, temperature distribution in the rat’s brain, electrical signals of neurons, etc. So, I recommend the authors provide these details to prove their micro-probe’s function before the publication of manuscript in Sensors.
A: A: The main purpose of the present manuscript is to share a set of novel processes for fabricating a new family of freestanding microprobes. As the Reviewer mentioned, these new probes have excellent mechanical properties and can be applied to detect some thermal or electrical signals at deep levels of live animals. The current results may help researchers in related fields to consider feasibility of alternative technical approaches. The electrochemical impedance spectroscopy of the Pd electrodes and size of the electrodes have been given in the revised version, and detailed thermal and electrical measurements were ongoing. We wish we could publish the results soon.

Reviewer 3 Report
This manuscript describes the wafer-level fabrication of stainless steel-based millimetric probes for deep-level sensing in tissues. The main novelty is the use of s.s. as base material, which the authors claim to have superior elastic and biocompatibility properties compared to available alternatives. The fabrication process is rather simple and clearly explained. The device was fully fabricated, calibrated and tested in an anestethised rat, as shown in the supporting video.
Overall the manuscript is interesting and worth adding to the extensive literature on microprobes. We have only a few minor remarks.
1) At the end of the introduction (lines 61-64), the authors comment on the possiblity of using steel for long times in tissues as in acupuncture, and on the biocompatibility of Cr and Ni. Both need references though, particularly the second may be controversial (e.g. Ni seems not recommended for money coins exactly because of biocompatibility issues).
2) In lines 77-81 the authors justify the placement of the probes within the substrate, however the reasoning is not clear nor consistent to us. Why only some of the swords point from the center to the periphery? Please clarify.
3) In line 82, the probes are described as "sword" shape, however in the reminder the authors use "sward", which has a very different meaning. Please amend.
4) In lines 137-140, figure 3 is referenced instead of figure 2, and su-8 -> SU-8.
5) The author report a significant improvement in surface roughness thanks to the SU-8 layer. What problems created the native roughness of s.s. in particular? We understand this is about yield, the authors should comment more extensively on this.
6) Testing of the elasticity of the probes is convincing (as in the video) but still qualitative. Can the authors quantify the Young's modulus of the probes?
7) For the calibration, it is not clear how the authors know the actual oil temperature, since they seem to control only the temperature of the hot plate.
8) While overall fine, the English needs some more careful check throughout the manuscript.
Author Response
Answers to the questions of Reviewer # 3
This manuscript describes the wafer-level fabrication of stainless steel-based millimetric probes for deep-level sensing in tissues. The main novelty is the use of s.s. as base material, which the authors claim to have superior elastic and biocompatibility properties compared to available alternatives. The fabrication process is rather simple and clearly explained. The device was fully fabricated, calibrated and tested in an anestethised rat, as shown in the supporting video.
Overall the manuscript is interesting and worth adding to the extensive literature on microprobes. We have only a few minor remarks.
Q1. At the end of the introduction (lines 61-64), the authors comment on the possibility of using steel for long times in tissues as in acupuncture, and on the biocompatibility of Cr and Ni. Both need references though, particularly the second may be controversial (e.g. Ni seems not recommended for money coins exactly because of biocompatibility issues).
A1: The current s.s. substrate may be not the best biocompatible material. However, in our applications thermal measurements usually take only seconds to minutes. So the toxicity of the TFTC materials might not a crucial point in these applications.
Q2. In lines 77-81 the authors justify the placement of the probes within the substrate, however the reasoning is not clear nor consistent to us. Why only some of the swords point from the center to the periphery? Please clarify.
A3: The fabrication processes involved a spinning coating of several kinds of photoresists, which were in liquid status. Our arrangement of the sward direction was favorable to obtain optimized uniformity of the polymer layer (photoresist) in spinning coating, when the probe tips are pointing to the center.
Q3. In line 82, the probes are described as "sword" shape, however in the reminder the authors use "sward", which has a very different meaning. Please amend.
A15: We have made correction in the revised version.
Q4. In lines 137-140, figure 3 is referenced instead of figure 2, and su-8 > SU-8
A4: We have made correction in the revised version.
Q5. The author reports a significant improvement in surface roughness thanks to the SU-8 layer. What problems created the native roughness of s.s. in particular? We understand this is about yield, the authors should comment more extensively on this.
A5: The rough, original surface always caused broken beams in the TFTC sensors. The original roughness of stainless steel was resulted from mechanical polishing process at the supplier’s lab. It is possible to reduce the surface roughness by additional mechanical-chemical polishing, but the cost may increase remarkably.
Q6. Testing of the elasticity of the probes is convincing (as in the video) but still qualitative. Can the authors quantify the Young's modulus of the probes?
A6: Yes, we have analyzed the stress-strain curve of the probes. The stress required to bend the stainless steel probe at 90 degrees was found to be 8-95 MPa, similar to that for silicon-based Utah electrode array (e.g. 82 MPa). The data were added in the revised version.
Q7. For the calibration, it is not clear how the authors know the actual oil temperature, since they seem to control only the temperature of the hot plate.
A7: During the calibration process, we placed the oil container on the hot plate. The oil temperature was controlled by heating the hot plate, and monitored with a standard Type-K thermocouple. The tip of the probe was dipped into oil to maintain the same temperature of the oil. The cold ends of the TFTC under test was recorded by another standard Type-K thermocouple that was firmly fixed to the cold ends. For different devices, one need different calibration systems and techniques. For details, please kindly refer to our previous papers: D. H. Han, J. J. Xu, Z. H. Wang, N. N. Yang, X. Z. Li, Y. Y. Qian, G. Li, R. J. Dai, and S. Y. Xu*, “Penetrating effect of high-intensity infrared laser pulses through body tissues”, RSC Adv., 8, (2018) 32344; F. Yang, G. Li, J. M. Yang, Z. H. Wang, D. H. Han, F. J. Zheng, and S. Y. Xu*, “Measurement of local temperature increments induced by cultured HepG2 cells with micro-thermocouples in a thermally stabilized system”, Sci. Rep., 7 (2017) 1721 (1-11); D. H. Han, S. K. Zhou, Z. H. Wang, G. Li, F. Yang, and S. Y. Xu*, “To save half contact pads in 2D mapping of local temperatures with a thermocouple array”, RSC Adv. 7, (2017) 9100; X. Y. Huo, H. X. Liu*, Y. R. Liang, M. Q. Fu, W. Q. Sun, Q. Chen, and S. Y. Xu*, “A nano-stripe based sensor for temperature measurement at the submicrometer and nano scales”, Small, 19, (2014) 3869-3875; H. X. Liu, W. Q. Sun, and S. Y. Xu*, “An extremely simple thermocouple made of a single layer of metal”, Adv. Mater. 24, (2012) 3275–3279; W. Q. Sun, H. X. Liu, and S. Y. Xu*, “Key issues in temperature sensing at the microscale with a thermocouple array”, Adv. Mater. Res. 422, (2012) 29-34; H. X. Liu, W. Q. Sun, Q. Chen, and S. Y. Xu*, “Thin-film thermocouple array for time-resolved local temperature mapping”, IEEE Eletron. Dev. Lett. 32, (2011) 1606-1608.
Q8. While overall fine, the English needs some more careful check throughout the manuscript.
A8: In the revised version we have made corrections, polished the language and rewritten several paragraphs. We have also obtained help from a native speaker to double check the text.

Round  2
Reviewer 1 Report
All me previous questions have been addressed. I think the electrode impedance is quite bad compared to state of the art. However this is a consequence of the material and design choice. At least the updated manuscript clearly mentions this. 
I understand the authors did not have the capability to measure the flatness/bending of the probe. This might be a problem though. The bending/flatness is critical to truly identify where in the tissue the recordings are being made. If the probe is used to identify tumor cells, it will be critical to know exactly (within<100um) where the recordings were made. But I understand the authors do not have the capability to do these measurements.
Reviewer 2 Report
In the revised version of manuscript, the authors didn't address all my questions and the explanation is not enough. Since the authors did the surgery on Sprague-Dawley rat, but they didn’t show any results about thermal or electrical signals from the rat’s brain. The micro-probe did not show its multifunction other than its mechanical property during the surgery process. So, I recommend the authors provide the basic recording results of thermal and electrical signals from live animals to prove their multifunctional micro-probe as a bio-device rather than a non-biocompatible electronic device before the publication of manuscript in Sensors